# Short-Time Propagators and the Born–Jordan Quantization Rule

**DOI:** 10.3390/e20110869

**Published:** 2018-11-10

**Authors:** Maurice A. de Gosson

**Affiliations:** Faculty of Mathematics (NuHAG), University of Vienna, Oskar-Morgenstern-Platz 1, 1090 Vienna, Austria; maurice.de.gosson@univie.ac.at

**Keywords:** Born–Jordan quantization, short-time propagators, time-slicing, Van Vleck determinant

## Abstract

We have shown in previous work that the equivalence of the Heisenberg and Schrödinger pictures of quantum mechanics requires the use of the Born and Jordan quantization rules. In the present work we give further evidence that the Born–Jordan rule is the correct quantization scheme for quantum mechanics. For this purpose we use correct short-time approximations to the action functional, initially due to Makri and Miller, and show that these lead to the desired quantization of the classical Hamiltonian.

## 1. Motivation and Background

### 1.1. Weyl versus Born and Jordan

There have been several attempts in the literature to find the “right” quantization rule for observables using either algebraic or analytical techniques [1,2,3,4,5,6,7]. In a recent paper [8] we have analyzed the Heisenberg and Schrödinger pictures of quantum mechanics, and shown that if one postulates that both theories are equivalent, then one must use the Born–Jordan quantization rule
(1)$$(\mathrm{BJ})\;\;\;\;x^{m}p^{\ell }⟶\frac{1}{m+1}\sum_{k=0}^{m}{\hat{x}}^{k}{\hat{p}}^{\ell }{\hat{x}}^{m-k},$$
and *not* the Weyl rule (To be accurate, it was McCoy [9] who showed that Weyl’s quantization scheme leads to Formula (Equation 2)).
(2)(Weyl)xmpℓ⟶12m∑k=0mmkx^kp^ℓx^m−k
for monomial observables. The Born–Jordan and Weyl rules yield the same result only if m<2 or ℓ<2; for instance in both cases the quantization of the product xp is $\frac{1}{2}\left(\hat{x}\hat{p}+\hat{p}\hat{x}\right)$. One can also show that the product pf(x) is, for any smooth function *f* of position alone, given in both cases by the symmetric rule
$$pf\left(x\right)⟶\frac{1}{2}\left(\hat{p}f\left(x\right)+f\left(x\right)\hat{p}\right).$$
It follows that if *H* is a Hamiltonian of the type
$$H=\sum_{j=1}^{n}\frac{1}{2m_{j}}{\left(p_{j}-A_{j}\left(x\right)\right)}^{2}+V\left(x\right)$$
one can use either the Weyl or the Born–Jordan prescriptions to get the the corresponding quantum operator, which yields the familiar expression
$$\hat{H}=\sum_{j=1}^{n}\frac{1}{2m_{j}}{\left(-i\hbar \frac{\partial }{\partial x_{j}}-A\left(x\right)\right)}^{2}+V\left(x\right).$$
(See Section 3.3). Since this Hamiltonian is without doubt the one which most often occurs in quantum mechanics one could ask why one should bother about which is the “correct” quantization. It turns out that this question is just a little bit more than academic: There are simple physical observables which yield different quantizations in the Weyl and Born–Jordan schemes. One interesting example is that of the squared angular momentum: Writing r=(x,y,z) and $p=(p_{x},p_{y},p_{z})$ the square of the classical angular momentum
(3)$$\ell =\left(yp_{z}-zp_{y}\right)i+\left(zp_{x}-xp_{z}\right)j+\left(xp_{y}-yp_{x}\right)k$$
is the function $\ell^{2}=\ell_{x}^{2}+\ell_{y}^{2}+\ell_{z}^{2}$ where
(4)$$\ell_{x}^{2}=x^{2}p_{y}^{2}+y^{2}p_{x}^{2}-2xp_{x}yp_{y}$$
and so on. The Weyl quantization of $\ell_{x}^{2}$ is
(5)$${\left(\hat{\ell_{x}^{2}}\right)}_{W}={\hat{x}}^{2}{\hat{p}}_{y}^{2}+{\hat{x}}_{y}^{2}{\hat{p}}_{x}^{2}-\frac{1}{2}\left(\hat{x}{\hat{p}}_{x}+{\hat{p}}_{x}\hat{x}\right)\left(\hat{y}{\hat{p}}_{y}+{\hat{p}}_{y}\hat{y}\right)$$
while its Born–Jordan quantization is
(6)$${\left(\hat{\ell_{x}^{2}}\right)}_{\mathrm{BJ}}={\hat{x}}^{2}{\hat{p}}_{y}^{2}+{\hat{x}}_{y}^{2}{\hat{p}}_{x}^{2}-\frac{1}{2}\left(\hat{x}{\hat{p}}_{x}+{\hat{p}}_{x}\hat{x}\right)\left(\hat{y}{\hat{p}}_{y}+{\hat{p}}_{y}\hat{y}\right)-\frac{1}{6}\hbar^{2};$$
similar relations are obtained for $\ell_{y}^{2}$ and $\ell_{z}^{2}$ so that, in the end,
(7)$${\left(\hat{\ell^{2}}\right)}_{W}-{\left(\hat{\ell^{2}}\right)}_{\mathrm{BJ}}=\frac{1}{2}\hbar^{2}.$$
This discrepancy has been dubbed the “angular momentum dilemma ”[10]; in [11] we have discussed this apparent paradox and shown that it disappears if one systematically uses Born–Jordan quantization.

### 1.2. The Kerner and Sutcliffe Approach to Quantization

As we have proven in [8,12], Heisenberg’s matrix mechanics [13], as rigorously constructed by Born and Jordan in [14] and Born, Jordan, and Heisenberg in [15], explicitly requires the use of the quantization rule (Equation 1) to be mathematically consistent, a fact which apparently has escaped the attention of physicists, and philosophers or historians of Science. In the present paper, we will show that the Feynman path integral approach is another genuinely physical motivation for Born–Jordan quantization of arbitrary observables; it corrects previous unsuccessful attempts involving path integral arguments which *do not work* for a reason that will be explained. One of the most convincing of these attempts is the paper [16] by Kerner and Sutcliffe. Elaborating on previous work of Garrod [17] Kerner and Sutcliffe tried to justify the Born–Jordan rule as the unique possible quantization (see Steven Kauffmann’s [18,19] brilliant discussion of this work). Assuming that $\hat{H}$ is the quantization of some general Hamiltonian *H*, they write as is usual in the theory of the phase space Feynman integral the propagator as
(8)$$\langle x|e^{-\frac{i}{\hbar }\hat{H}t}|x^{\prime }\rangle =\lim_{N\to \infty }\int dx_{N-1}\cdots dx_{1}\prod_{k=1}^{N}\langle x_{k}|e^{-\frac{i}{\hbar }\hat{H}\Delta t}|x_{k-1}\rangle$$
where $x_{N}=x$ and $x_{0}=x^{\prime }$ are fixed and Δt=t/N. They thereafter use the approximation
(9)$$\langle x_{k}|e^{-\frac{i}{\hbar }\hat{H}\Delta t}|x_{k-1}\rangle \approx \frac{1}{2\pi \hbar }\int e^{\frac{i}{\hbar }\bar{S}\left(x,x^{\prime },p,\Delta t\right)}dp$$
the function $\bar{S}$ being given by
(10)$$\bar{S}\left(x,x^{\prime },p,\Delta t\right)=p\left(x-x^{\prime }\right)-\bar{H}\left(x,x^{\prime },p\right)\Delta t$$
where $\bar{H}$ is the time average of *H* over *p* fixed and x=x(t), that is
(11)$$\bar{H}\left(x,x^{\prime },p\right)=\frac{1}{\Delta t}\int_{0}^{\Delta t}H\left(x^{\prime }+s\frac{x-x^{\prime }}{\Delta t},p\right)ds.$$
Notice that introducing the dimensionless parameter τ=s/Δt, Formula (Equation 11) can be written in the more convenient form
(12)$$\bar{H}\left(x,x^{\prime },p\right)=\int_{0}^{1}H\left(\tau x+\left(1-\tau \right)x^{\prime },p\right)d\tau$$
which is the usual mathematical definition of Born–Jordan quantization: See de Gosson [12,20] and de Gosson and Luef [21].

Taking the limit Δt→0 the operator $\hat{H}$ can then be explicitly and uniquely determined, and Kerner and Sutcliffe show that in particular this leads to the Born–Jordan ordering (Equation 1) when their Hamiltonian *H* is a monomial $x^{m}p^{\ell }$. Unfortunately (as immediately Cohen’s rebuttal was published in the same volume of *J. Math. Phys.* in which Kerner and Sutcliffe published their results. Noted by Cohen [22]) there are many a priori equally good constructions of the Feynman integral, leading to other quantization rules. In fact, argues Cohen, there is a great freedom of choice in calculating the action $p\left(x-x^{\prime }\right)-\bar{H}$ appearing in the right-hand side of (Equation 11). For instance, one can choose
(13)$$S\left(x,x^{\prime },p,\Delta t\right)=p\left(x-x^{\prime }\right)-H\left(\frac{1}{2}\left(x+x^{\prime }\right),p\right)\Delta t$$
which leads for $x^{m}p^{\ell }$ to Weyl’s rule (Equation 2), or one can choose
(14)$$S\left(x,x^{\prime },p,\Delta t\right)=p\left(x-x^{\prime }\right)-\frac{1}{2}\left(H\left(x,p\right)+H\left(x^{\prime },p\right)\right)\Delta t,$$
which leads to the symmetric rule
(15)$$x^{m}p^{\ell }⟶\frac{1}{2}\left({\hat{x}}^{m}{\hat{p}}^{\ell }+{\hat{p}}^{\ell }{\hat{x}}^{m}\right).$$

This ambiguity shows—in an obvious way—that Feynman path integral theory does not lead to an uniquely defined quantization scheme for observables. However—and this is the main point of the present paper—while Cohen’s remark was mathematically justified, Kerner and Sutcliffe’s insight was right (albeit for the wrong reason).

### 1.3. What We Will Do

It turns out that the Formula (Equation 10) for the approximate action that Kerner and Sutcliffe “guessed” has been justified independently (in another context) by Makri and Miller [23,24] and the present author [25] by rigorous mathematical methods. This formula is actually the *correct* approximation to action up to order $O(\Delta t^{2})$ (as opposed to the “midpoint rules” commonly used in the theory of the Feynman integral which yield much cruder approximations); it follows that Kerner and Sutcliffe’s Formula (Equation 9) indeed yields a correct approximation of the infinitesimal propagator $\langle x_{k}|e^{-\frac{i}{\hbar }\hat{H}\Delta t}|x_{k-1}\rangle$, in fact the *best one* for calculational purposes since it ensures a swift convergence of numerical schemes. This is because for short times Δt the solution of Schrödinger’s equation
(16)$$i\hbar \frac{\partial \psi }{\partial t}\left(x,t\right)=\left[\sum_{j=1}^{n}\frac{-\hbar^{2}}{2m_{j}}\frac{\partial^{2}}{\partial x_{j}^{2}}+V\left(x\right)\right]\psi \left(x,t\right)$$
with initial condition $\psi \left(x,0\right)=\psi_{0}\left(x\right)$ is given by the asymptotic formula
(17)$$\psi \left(x,\Delta t\right)=\int \bar{K}\left(x,x^{\prime },\Delta t\right)\psi_{0}\left(x^{\prime }\right)d^{n}x^{\prime }+O\left(\Delta t^{2}\right);$$
the approximate propagator $\bar{K}$ being defined, for arbitrary time *t*, by
(18)$$\bar{K}\left(x,x^{\prime },t\right)={\left(\frac{1}{2\pi \hbar }\right)}^{n}\int \exp\left(\frac{i}{\hbar }\left[p\left(x-x^{\prime }\right)-\left(H_{\mathrm{free}}\left(p\right)+\bar{V}\left(x,x^{\prime }\right)\right)t\right]\right)d^{n}p,$$
where, by definition, $H_{\mathrm{free}}\left(p\right)$ is the free particle Hamiltonian function, and the two-point function
$$\bar{V}\left(x,x^{\prime }\right)=\int_{0}^{1}V\left(\tau x+\left(1-\tau \right)x^{\prime }\right)d\tau$$
is the average value of the potential *V* on the line segment $\left[x^{\prime },x\right]$.
In Section 2 we discuss the accuracy of Kerner and Sutcliffe’s propagator by comparing it with the more familiar Van Vleck propagator; we show that for small times both are approximations to order $O(t^{2})$ to the exact propagator of Schrödinger’s equation.In Section 3 we show that if one assume’s that short-time evolution of the wavefunction (for an arbitrary Hamiltonian *H*) is given by the Kerner and Sutcliffe propagator, then *H* must be quantized following the rule (Equation 12); we thereafter show that when *H* is a monomial $x^{m}p^{\ell }$ then the corresponding operator is given by the Born–Jordan rule (Equation 1), *not* by the Weyl rule Equation 2.


**Notation** **1.**
*The generalized position and momentum vectors are $x=(x_{1},\ldots ,x_{n})$ and $p=(p_{1},\ldots ,p_{n})$; we set $px=p_{1}x_{1}+\cdots +p_{n}x_{n}$. We denote by ${\hat{x}}_{j}$ the operator of multiplication by $x_{j}$ and by ${\hat{p}}_{j}$ the momentum operator $-i\hbar (\partial /\partial x_{j})$.*


## 2. On Short-Time Propagators

In this section we only consider Hamiltonian functions of the type “kinetic energy plus potential”:(19)$$H\left(x,p\right)=H_{\mathrm{free}}\left(p\right)+V\left(x\right)\;\;,\;\;H_{\mathrm{free}}\left(p\right)=\sum_{j=1}^{n}\frac{1}{2m_{j}}p_{j}^{2}.$$
These are the simplest physical Hamiltonians, both from a classical and a quantum perspective.

### 2.1. The Van Vleck Propagator

Consider a Hamiltonian function of the type (Equation 19) above; the corresponding Schrödinger equation is
(20)$$i\hbar \frac{\partial \psi }{\partial t}\left(x,t\right)=\left[\sum_{j=1}^{n}\frac{-\hbar^{2}}{2m_{j}}\frac{\partial^{2}}{\partial x_{j}^{2}}+V\left(x\right)\right]\psi \left(x,t\right).$$
We will denote by $K\left(x,x^{\prime },t\right)=\langle x|e^{-\frac{i}{\hbar }\hat{H}t}|x^{\prime }\rangle$ the corresponding exact propagator:(21)$$\psi \left(x,t\right)=\int K\left(x,x^{\prime },t\right)\psi_{0}\left(x^{\prime }\right)d^{n}x^{\prime }$$
where with $\psi_{0}\left(x\right)$ is the value of ψ at time t=0. The function $K(x,x^{\prime },t)$ must thus satisfy the boundary condition
(22)$$\lim_{t\to 0}K\left(x,x^{\prime },t\right)=\delta \left(x-x^{\prime }\right).$$

It is well-known (see e.g., Gutzwiller [26], Schulman [27], de Gosson [25], Maslov and Fedoriuk [28]) that for short times an approximate propagator is given by Van Vleck’s formula
(23)$$\tilde{K}\left(x,x^{\prime },t\right)={\left(\frac{1}{2\pi i\hbar }\right)}^{n/2}\sqrt{\rho (x,x^{\prime },t)}e^{\frac{i}{\hbar }S\left(x,x^{\prime },t\right)}$$
where
(24)$$S\left(x,x^{\prime },t\right)=\int_{0}^{t}\left(\sum_{j=1}^{n}\frac{1}{2}m_{j}{\dot{x}}_{j}{\left(s\right)}^{2}-V(x\left(s\right)\right)ds$$
is the action along the classical trajectory leading from $x^{\prime }$ at time $t^{\prime }=0$ to *x* at time *t* (there is no sum over different classical trajectories because only one trajectory contributes in the limit t→0 [23]) and
(25)$$\rho \left(x,x^{\prime },t\right)=\det{\left(-\frac{\partial^{2}S\left(x,x^{\prime },t\right)}{\partial x_{j}\partial x_{jk}^{\prime }}\right)}_{1\leq j,k\leq n}$$
is the Van Vleck density of trajectories [25,26,27]; the argument of the square root is chosen so that the initial condition (Equation 22) is satisfied [25,29]. It should be emphasized that although the Van Vleck propagator is frequently used in semiclassical mechanics, it has nothing “semiclassical” *per se*, since it is genuinely an approximation to the exact propagator for small *t* – not just in the limit ℏ→0. In fact:
**Theorem** **1.***Let $\tilde{\psi }$ be given by*$$\tilde{\psi }\left(x,t\right)=\int \tilde{K}\left(x,x^{\prime },t\right)\psi_{0}\left(x^{\prime }\right)d^{n}x^{\prime }$$*where $\psi_{0}$ is a tempered distribution. Let ψ be the exact solution of Schrödinger’s equation with initial datum $\psi_{0}$. We have*(26)$$\psi \left(x,t\right)-\tilde{\psi }\left(x,t\right)=O\left(t^{2}\right).$$*In particular, the Van Vleck propagator $\tilde{K}\left(x,x^{\prime },t\right)$ is an $O(t^{2})$ approximation to the exact propagator $K(x,x^{\prime },t)$:*(27)$$K\left(x,x^{\prime },t\right)-\tilde{K}\left(x,x^{\prime },t\right)=O\left(t^{2}\right)$$*for t→0 and hence*$$\lim_{t\to 0}\tilde{K}\left(x,x^{\prime },t\right)=\delta \left(x-x^{\prime }\right).$$
**Proof.** Referring to de Gosson [25] (Lemma 241) for details, we sketch the main lines in the case n=1. Assuming that $\psi_{0}$ belongs to the Schwartz space $S(R^{n})$ of rapidly decreasing functions, one expands the solution ψ of Schrödinger’s equation to second order:
$$\psi \left(x,t\right)=\psi_{0}\left(x\right)+\frac{\partial \psi }{\partial t}\left(x,0\right)t+O\left(t^{2}\right).$$
Taking into account the fact that ψ is a solution of Schrödinger’s equation this can be rewritten
(28)$$\psi \left(x,t\right)=\left[1+\frac{t}{i\hbar }\left(-\frac{\hbar^{2}}{2m}\frac{\partial^{2}}{\partial x^{2}}+V\left(x\right)\right)\right]\psi_{0}\left(x\right)+O\left(t^{2}\right).$$
Expanding the exponential $e^{iS/\hbar }$ in Van Vleck’s Formula (Equation 23) at t=0 one shows, using the estimate (Equation 32) in Theorem 2, that we also have
(29)$$\tilde{\psi }\left(x,t\right)=\left[1+\frac{t}{i\hbar }\left(-\frac{\hbar^{2}}{2m}\frac{\partial^{2}}{\partial x^{2}}+V\left(x\right)\right)\right]\psi_{0}\left(x\right)+O\left(t^{2}\right);$$
comparison with (Equation 28) implies that $\psi \left(x,t\right)-\tilde{\psi }\left(x,t\right)=O\left(t^{2}\right)$. By density of the Schwartz space in the class of tempered distributions $S^{\prime }\left(R^{n}\right)$ the estimate (Equation 26) is valid if one chooses $\psi_{0}\left(x\right)=\delta \left(x-x_{0}\right)$, which yields Formula (Equation 27) since we have
$$\int \tilde{K}\left(x,x^{\prime },t\right)\delta \left(x-x_{0}\right)d^{n}x^{\prime }=\tilde{K}\left(x,x_{0},t\right)$$
and
$$\int K\left(x,x^{\prime },t\right)\delta \left(x-x_{0}\right)d^{n}x^{\prime }=K\left(x,x_{0},t\right).$$ □



Let us briefly return to the path integral. Replacing the terms $\langle x_{k}|e^{-\frac{i}{\hbar }\hat{H}\Delta t}|x_{k-1}\rangle$ in the product Formula (Equation 8) with $\tilde{K}\left(x_{k-1},x_{k-1},\Delta t\right)$ one shows, using the Lie–Trotter Formula [25,27], that the exact propagator $K\left(x,x^{\prime },t\right)=\langle x|e^{-\frac{i}{\hbar }\hat{H}t}|x^{\prime }\rangle$ is given by
(30)$$\langle x|e^{-\frac{i}{\hbar }\hat{H}t}|x^{\prime }\rangle =\lim_{N\to \infty }\int dx_{N-1}\cdots dx_{1}\prod_{k=1}^{N}\tilde{K}\left(x_{k-1},x_{k-1},\Delta t\right).$$
This formula is often taken as the starting point of path integral arguments: observing that the expression (Equation 23) is in most cases (The free particle and the harmonic oscillator are remarkable particular cases where the action integral can be explicitly calculated and thus yields an explicit formula for the propagator, but mathematically speaking this fact is rather a consequence of the theory of the metaplectic group [25,29]) difficult to calculate (it implies the computation of an action integral, which can be quite cumbersome) people working in the theory of the Feynman integral replace the exact action $S(x,x^{\prime },t)$ in (Equation 23) with approximate expressions, for instance the “midpoint rules” that will be discussed below. Now, one should be aware that this *legerdemain* works, because when taking the limit N→∞ one indeed obtains the correct propagator, but it does *not* imply that these midpoint rules are accurate approximations to $S(x,x^{\prime },t)$.

### 2.2. The Kerner–Sutcliffe Propagator

We showed above that the Van Vleck propagator is an approximation to order $O(t^{2})$ to the exact propagator. We now show that the propagator proposed by Kerner and Sutcliffe in [16] approximates the Van Vleck propagator also at order $O(t^{2})$. Hence
$$\mathrm{Van}\;\mathrm{Vleck}=\mathrm{Kerner}-\mathrm{Sutcliffe}+O(t^{2}).$$
We begin by giving a correct short-time approximation to the action.

**Theorem** **2.**
*The function $\bar{S}$ defined by*
(31)$$\bar{S}\left(x,x^{\prime },t\right)=\sum_{j=1}^{n}m_{j}\frac{{\left(x_{j}-x_{j}^{\prime }\right)}^{2}}{2t}-\bar{V}\left(x,x^{\prime }\right)t$$
*where $\bar{V}\left(x,x^{\prime }\right)$ is the average of the potential V along the line segment $[x^{\prime },x]:$*
$$\bar{V}\left(x,x^{\prime }\right)=\int_{0}^{1}V\left(\tau x+\left(1-\tau \right)x^{\prime }\right)d\tau .$$
*satisfies for t→0 the estimate*
(32)$$S\left(x,x^{\prime },t\right)-\bar{S}\left(x,x^{\prime },t\right)=O\left(t^{2}\right).$$


For detailed proofs we refer to the aforementioned papers [23,24] by Makri and Miller, and to our book [25]; also see de Gosson and Hiley [30,31]. The underlying idea is quite simple (and already appears in germ in Park’s book [32], p. 438): one remarks that the function $S=S(x,x^{\prime },t)$ satisfies the Hamilton–Jacobi equation
(33)$$\frac{\partial S}{\partial t}+\sum_{j=1}^{n}\frac{1}{2m_{j}}{\left(\frac{\partial S}{\partial x_{j}}\right)}^{2}+V\left(x\right)=0$$
and one thereafter looks for an asymptotic solution
$$S\left(x,x^{\prime },t\right)=\frac{1}{t}S_{0}\left(x,x^{\prime }\right)+S_{1}\left(x,x^{\prime }\right)t+S_{2}\left(x,x^{\prime }\right)t^{2}+\cdots .$$
Insertion in (Equation 33) then leads to
$$S_{0}\left(x,x^{\prime }\right)=\sum_{j=1}^{n}m_{j}\frac{{\left(x_{j}-x_{j}^{\prime }\right)}^{2}}{2}$$
and $S_{1}\left(x,x^{\prime }\right)=-\bar{V}\left(x,x^{\prime }\right)$ hence (Equation 31). Notice that this procedure actually allows one to find approximations to *S* to an arbitrary order of accuracy by solving successively the equations satisfied by $S_{2}$, $S_{3},\ldots$ (see [23,24] for explicit formulas).

Let us now set
$$\bar{H}\left(x,x^{\prime },t\right)=H_{\mathrm{free}}\left(p\right)+\bar{V}\left(x,x^{\prime }\right)$$
where
$$\bar{V}\left(x,x^{\prime }\right)=\int_{0}^{1}V\left(\tau x+\left(1-\tau \right)x^{\prime }\right)d\tau$$
is the averaged potential.

Let us now show that the propagator postulated by Garrod [17] and Kerner and Sutcliffe [16] is as good an approximation to the exact propagator as Van Vleck’s is. We recall the textbook Fourier formula
(34)$${\left(\frac{1}{2\pi \hbar }\right)}^{n}\int e^{\frac{i}{\hbar }p\left(x-x^{\prime }\right)}p_{j}^{\ell }d^{n}p={\left(-i\hbar \frac{\partial }{\partial x_{j}}\right)}^{\ell }\delta \left(x-x^{\prime }\right).$$

**Theorem** **3.**
*Let $\bar{K}=\bar{K}\left(x,x^{\prime },t\right)$ be defined (in the distributional sense) by*
(35)$$\bar{K}\left(x,x^{\prime },t\right)={\left(\frac{1}{2\pi \hbar }\right)}^{n}\int e^{\frac{i}{\hbar }\left(p\left(x-x^{\prime }\right)-\bar{H}\left(x,x^{\prime },p\right)t\right)}d^{n}p.$$
*and set*
(36)$$\bar{\psi }\left(x,t\right)=\int \bar{K}\left(x,x^{\prime },t\right)\psi_{0}\left(x^{\prime }\right)d^{n}x^{\prime }.$$
*Let ψ be the solution of Schrödinger’s equation with initial condition $\psi_{0}$. We have*
(37)$$\bar{\psi }\left(x,t\right)-\psi \left(x,t\right)=O\left(t^{2}\right).$$
*The function $\bar{K}$ is an $O(t^{2})$ approximation to the exact propagator K:*
(38)$$K\left(x,x^{\prime },t\right)-\bar{K}\left(x,x^{\prime },t\right)=O\left(t^{2}\right).$$


**Proof.** It is sufficient to prove (Equation 37); Formula (Equation 38) follows by the same argument as in the proof of Theorem 1. To simplify notation we assume again n=1; the general case is a straightforward extension. Expanding for small *t* the exponential in the integrand of (Equation 35) we have
$$\begin{array}{cc} \bar{K}\left(x,x^{\prime },t\right) & ={\left(\frac{1}{2\pi \hbar }\right)}^{n}\int e^{\frac{i}{\hbar }p\left(x-x^{\prime }\right)}\left(1-\frac{i}{\hbar }\bar{H}\left(x,x^{\prime },p\right)t\right)dp+O\left(t^{2}\right)\\ & =\delta \left(x-x^{\prime }\right)-\frac{it}{\hbar }\int e^{\frac{i}{\hbar }p\left(x-x^{\prime }\right)}\bar{H}\left(x,x^{\prime },p\right)dp+O\left(t^{2}\right) \end{array}$$
and hence
$$\bar{\psi }\left(x,t\right)=\psi_{0}\left(x\right)-\frac{it}{\hbar }\int e^{\frac{i}{\hbar }p\left(x-x^{\prime }\right)}\bar{H}\left(x,x^{\prime },p\right)\psi_{0}\left(x^{\prime }\right)dpdx^{\prime }+O\left(t^{2}\right).$$
We have
$$\int e^{\frac{i}{\hbar }p\left(x-x^{\prime }\right)}\bar{H}\left(x,x^{\prime },p\right)d^{n}p=\int e^{\frac{i}{\hbar }p\left(x-x^{\prime }\right)}\left(\frac{p^{2}}{2m}+\bar{V}\left(x,x^{\prime }\right)\right)dp;$$
using the Fourier Formula (Equation 34) we get
$${\left(\frac{1}{2\pi \hbar }\right)}^{n}\int e^{\frac{i}{\hbar }p\left(x-x^{\prime }\right)}\frac{p^{2}}{2m}dp=-\frac{\hbar^{2}}{2m}\frac{\partial^{2}}{\partial x^{2}}\delta \left(x-x^{\prime }\right)$$
and, noting that $\bar{V}\left(x,x\right)=V\left(x\right)$,
$$\begin{array}{cc} {\left(\frac{1}{2\pi \hbar }\right)}^{n}\int e^{\frac{i}{\hbar }p\left(x-x^{\prime }\right)}\bar{V}\left(x,x^{\prime }\right)dp & =\bar{V}\left(x,x^{\prime }\right)\delta \left(x-x^{\prime }\right)\\ & =V\left(x\right)\delta \left(x-x^{\prime }\right). \end{array}$$
Summarizing,
(39)$$\bar{K}\left(x,x^{\prime },t\right)=\delta \left(x-x^{\prime }\right)+\frac{it}{\hbar }\left(-\frac{\hbar^{2}}{2m}\frac{\partial^{2}}{\partial x^{2}}+V\left(x\right)\right)\delta \left(x-x^{\prime }\right)+O\left(t^{2}\right)$$
and hence
$$\bar{\psi }\left(x,t\right)=\psi_{0}\left(x\right)-\frac{it}{\hbar }\left(-\frac{\hbar^{2}}{2m}\frac{\partial^{2}}{\partial x^{2}}+V\left(x\right)\right)\psi_{0}\left(x\right)+O\left(t^{2}\right).$$
Comparing this expression with (Equation 28) yields (Equation 38). □

### 2.3. Comparison of Short-Time Propagators

We have seen above that both the Van Vleck and the Kerner–Sutcliffe propagators are accurate to order $O(t^{2})$:(40)$$\begin{array}{cc} K\left(x,x^{\prime },t\right)-\tilde{K}\left(x,x^{\prime },t\right) & =O(t^{2}). \end{array}$$(41)$$\begin{array}{cc} K\left(x,x^{\prime },t\right)-\bar{K}\left(x,x^{\prime },t\right) & =O(t^{2}) \end{array}$$
and hence, of course,
(42)$$\tilde{K}\left(x,x^{\prime },t\right)-\bar{K}\left(x,x^{\prime },t\right)=O\left(t^{2}\right).$$
Let us now study the case of the most commonly approximations to the action used in the theory of the Feynman integral, namely the mid-point rules
(43)$$S_{1}\left(x,x^{\prime },t,t^{\prime }\right)=\sum_{j=1}^{n}m_{j}\frac{{\left(x_{j}-x_{j}^{\prime }\right)}^{2}}{2t}-\frac{1}{2}\left(V\left(x\right)+V\left(x^{\prime }\right)\right)t$$
and
(44)$$S_{2}\left(x,x^{\prime },t\right)=\sum_{j=1}^{n}m_{j}\frac{{\left(x_{j}-x_{j}^{\prime }\right)}^{2}}{2t}-V\left(\frac{1}{2}\left(x+x^{\prime }\right)\right)\Delta t.$$
We begin with a simple example, that of the harmonic oscillator
$$H\left(x,p\right)=\frac{p^{2}}{2m}+\frac{1}{2}m^{2}\omega^{2}x^{2}$$
(we are assuming n=1). The exact value of the action is given by the generating function
(45)$$S\left(x,x^{\prime },t\right)=\frac{m}{2\sin\omega t}\left(\left(x^{2}+x^{\prime 2}\right)\cos\omega t-2xx^{\prime }\right);$$
expanding the terms sinωt and cosωt in Taylor series for t→0 yields the approximation
(46)$$S\left(x,x^{\prime },t\right)=m\frac{{\left(x-x^{\prime }\right)}^{2}}{2t}-\frac{m\omega^{2}}{6}\left(x^{2}+xx^{\prime }+x^{\prime 2}\right)t+O\left(t^{2}\right).$$
It is easy to verify, averaging $\frac{1}{2}m^{2}\omega^{2}x^{2}$ over $\left[x^{\prime },x\right]$ that
$$\bar{S}\left(x,x^{\prime },t\right)=m\frac{{\left(x-x^{\prime }\right)}^{2}}{2t}-\frac{m\omega^{2}}{6}\left(x^{2}+xx^{\prime }+x^{\prime 2}\right)t$$
is precisely the approximate action provided by (Equation 31). If we now instead apply the midpoint rule (Equation 43) we get
$$S_{1}\left(x,x^{\prime },t\right)=m\frac{{\left(x-x^{\prime }\right)}^{2}}{2t}-\frac{m^{2}\omega^{2}}{4}\left(x^{2}+x^{\prime 2}\right)t$$
which differs from the correct value (Equation 46) by a term O(Δt). Similarly, the rule (Equation 44) yields
$$S_{2}\left(x,x^{\prime },t\right)=m\frac{{\left(x-x^{\prime }\right)}^{2}}{2t}-\frac{m^{2}\omega^{2}}{8}{\left(x+x^{\prime }\right)}^{2}t$$
which again differs from the correct value (Equation 45) by a term O(t). It is easy to understand why it is so by examining the case of a general potential function, and to compare $\bar{V}\left(x,x^{\prime }\right)$, $\frac{1}{2}\left(V\left(x\right)+V\left(x^{\prime }\right)\right)$, and $V(\frac{1}{2}\left(x+x^{\prime }\right)$. Consider for instance $\bar{V}\left(x,x^{\prime }\right)-V(\frac{1}{2}\left(x+x^{\prime }\right)$. Expanding V(x) in a Taylor series at $\bar{x}=\frac{1}{2}\left(x+x^{\prime }\right)$ we get after some easy calculations
$$\begin{array}{cc} \bar{V}\left(x,x^{\prime }\right) & =V\left(\bar{x}\right)+V^{\prime }\left(\bar{x}\right)\left(x-x^{\prime }\right)+\frac{1}{2}V^{\prime \prime }\left(\bar{x}\right){\left(x-x^{\prime }\right)}^{2}+O\left({\left(x-x^{\prime }\right)}^{3}\right)\\ & =V(\frac{1}{2}\left(x+x^{\prime }\right)-\frac{1}{12}V^{\prime \prime }\left(\frac{1}{2}\left(x+x^{\prime }\right)\right){\left(x-x^{\prime }\right)}^{3}+O\left({\left(x-x^{\prime }\right)}^{3}\right) \end{array}$$
hence $\bar{V}\left(x,x^{\prime }\right)-V(\frac{1}{2}\left(x+x^{\prime }\right)$ is different from zero unless $x=x^{\prime }$ (or if V(x) is linear) and hence the difference between $\bar{S}\left(x,x^{\prime },t\right)$ and $S_{2}\left(x,x^{\prime },t\right)$ will always generate a term containing *t* so that $\bar{S}\left(x,x^{\prime },t\right)-S_{2}\left(x,x^{\prime },t\right)=O\left(t\right)$ (and not $O(t^{2})$). A similar calculation shows that we will also always have $\bar{S}\left(x,x^{\prime },t\right)-S_{1}\left(x,x^{\prime },t\right)=O\left(t\right)$. Denoting by $K_{1}\left(x,x^{\prime },t\right)$ and $K_{2}\left(x,x^{\prime },t\right)$ the approximate propagators obtained from the midpoint rules (Equation 43) and (Equation 44), respectively, one checks without difficulty that we will have
$$\begin{array}{cc} \bar{K}\left(x,x^{\prime },t\right)-K_{1}\left(x,x^{\prime },t\right) & =O(t)\\ \bar{K}\left(x,x^{\prime },t\right)-K_{2}\left(x,x^{\prime },t\right) & =O(t) \end{array}$$
where $\bar{K}\left(x,x^{\prime },t\right)$ is the Kerner–Sutcliffe propagator (Equation 35) (in these relations we can of course replace $\bar{K}\left(x,x^{\prime },t\right)$ with the van Vleck propagator $\tilde{K}\left(x,x^{\prime },t\right)$ since both differ by a quantity $O(t^{2})$ in view of Theorem 3.

## 3. The Case of Arbitrary Hamiltonians

### 3.1. The Main Result

We now consider the following very general situation: We assume that we are in the presence of a quantum system represented by a state |ψ〉 whose evolution is governed by a strongly continuous one-parameter group $\left(U_{t}\right)$ of unitary operators acting on $L^{2}\left(R^{n}\right)$; the operator $U_{t}$ takes an initial wavefunction $\psi_{0}$ to $\psi =U_{t}\psi_{0}$. It follows from Schwartz’s kernel theorem [33] that there exists a function $K=K(x,x^{\prime };t)$ such that (This equality is sometimes postulated; it is in fact a mathematical fact which is true in quite general situations.)
(47)$$\psi \left(x,t\right)=\int K\left(x,x^{\prime };t\right)\psi_{0}\left(x^{\prime }\right)d^{n}x^{\prime }$$
and from Stone’s [34] theorem one strongly continuous one-parameter groups of unitary operators that there exists a self-adjoint (generally unbounded) operator $\hat{H}$ on $L^{2}\left(R^{n}\right)$ such that
(48)$$\psi \left(x,t\right)=e^{-\frac{i}{\hbar }\hat{H}t}\psi_{0}\left(x\right);$$
equivalently ψ(x,t) satisfies the abstract Schrödinger equation (Jauch [35])
(49)$$i\hbar \frac{\partial \psi }{\partial t}\left(x,t\right)=\hat{H}\psi \left(x,t\right).$$

We now make the following crucial assumption, which extrapolates to the general case what we have done for Hamiltonians of the type classical type “kinetic energy plus potential”: the quantum dynamics is again given by the Kerner–Sutcliffe propagator (Equation 35) for small times *t*, i.e.,
(50)$$K\left(x,x^{\prime },t\right)=\bar{K}\left(x,x^{\prime },t\right)+O\left(t^{2}\right)$$
the approximate propagator being given by
(51)$$\bar{K}\left(x,x^{\prime },t\right)={\left(\frac{1}{2\pi \hbar }\right)}^{n}\int e^{\frac{i}{\hbar }\left(p\left(x-x^{\prime }\right)-\bar{H}\left(x,x^{\prime }\right)t\right)}d^{n}p$$
where $\bar{H}$ is this time the averaged Hamiltonian function
(52)$$\bar{H}\left(x,x^{\prime },p\right)=\int_{0}^{1}H\left(\tau x+\left(1-\tau \right)x^{\prime },p\right)d\tau .$$
Obviously, when $H=H_{\mathrm{free}}+V$ the function $\bar{H}$ reduces to the function $H_{\mathrm{free}}+\bar{V}$ considered in Section 2.

This assumption can be motivated as follows (see de Gosson [12], Proposition 15, §4.4). Let
$$S\left(x,x^{\prime },t\right)=\int_{\gamma }pdx-Hdt$$
be Hamilton’s two-point function calculated along the phase space path leading from an initial point $\left(x^{\prime },p^{\prime },0\right)$ to a final point (x,p,t) (the existence of such a function for small *t* is guaranteed by Hamilton–Jacobi theory; see e.g., Arnol’d [36] or Goldstein [37]). That function satisfies the Hamilton–Jacobi equation
$$\frac{\partial S}{\partial t}+H\left(x,\nabla_{x}S\right)=0.$$
One then shows that the function
$$\bar{S}\left(x,x^{\prime },t\right)=p\left(x-x^{\prime }\right)-\bar{H}\left(x,x^{\prime },p\right)t$$
where *p* is the momentum at time *t* is an approximation to $S(x,x^{\prime },t)$, in fact
$$\bar{S}\left(x,x^{\prime },t\right)-S\left(x,x^{\prime },t\right)=O\left(t^{2}\right).$$
Here is an example: Choose $H=\frac{1}{2}p^{2}x^{2}$ (we are assuming here n=1); then
$$S\left(x,x^{\prime },t\right)=\frac{{\left(\ln\left(x/x^{\prime }\right)\right)}^{2}}{2t}.$$
Using the formula
$$\bar{H}\left(x,x^{\prime },p\right)=\frac{1}{6}p^{2}\left(x^{2}+xx^{\prime }+x^{\prime 2}\right)$$
one shows after some calculations involving the Hamiltonian equations for *H* that
$$\bar{S}\left(x,x^{\prime },t\right)=\frac{{\left(\ln\left(x/x^{\prime }\right)\right)}^{2}}{2t}+O\left(t^{2}\right)$$
(see [12], Chapter 4, Examples 10 and 16 for detailed calculations).

We are now going to show that the operator $\hat{H}$ can be explicitly and uniquely determined from the knowledge of $\bar{K}\left(x,x^{\prime },t\right)$.

**Theorem** **4.**
*If we assume that the short-time propagator is given by formula (Equation 51) then the operator $\hat{H}$ appearing in the abstract Schrödinger Equation (Equation 49) is given by*
(53)$$\hat{H}\psi \left(x\right)={\left(\frac{1}{2\pi \hbar }\right)}^{n}\int e^{\frac{i}{\hbar }p\left(x-x^{\prime }\right)}\bar{H}\left(x,x^{\prime },p\right)\psi \left(x^{\prime }\right)d^{n}pd^{n}x^{\prime }.$$


**Proof.** Differentiating both sides of the equality (Equation 47) with respect to time we get
$$i\hbar \frac{\partial \psi }{\partial t}\left(x,t\right)=i\hbar \int \frac{\partial K}{\partial t}\left(x,x^{\prime },t\right)\psi_{0}\left(x^{\prime }\right)d^{n}x^{\prime };$$
since *K* itself satisfies the Schrödinger Equation (Equation 49) we thus have
$$\hat{H}\psi \left(x,t\right)=i\hbar \int \frac{\partial K}{\partial t}\left(x,x^{\prime },t\right)\psi_{0}\left(x^{\prime }\right)d^{n}x^{\prime }.$$
It follows, using the assumptions (Equation 50) and (Equation 51), that
$$\hat{H}\psi \left(x,t\right)=i\hbar \int \frac{\partial \bar{K}}{\partial t}\left(x,x^{\prime },t\right)\psi_{0}\left(x^{\prime }\right)d^{n}x^{\prime }+O\left(t\right)$$
and hence, letting t→0,
(54)$$\hat{H}\psi_{0}\left(x\right)=i\hbar \int \frac{\partial \bar{K}}{\partial t}\left(x,x^{\prime },0\right)\psi_{0}\left(x^{\prime }\right)d^{n}x^{\prime }.$$
Introducing the notation
$$\bar{S}\left(x,x^{\prime },t\right)=p\left(x-x^{\prime }\right)-\bar{H}\left(x,x^{\prime },p\right)t$$
we have
$$\begin{array}{cc} \frac{\partial \bar{K}}{\partial t}\left(x,x^{\prime },t\right) & ={\left(\frac{1}{2\pi \hbar }\right)}^{n}\frac{i}{\hbar }\int e^{\frac{i}{\hbar }\bar{S}\left(x,x^{\prime },t\right)}\frac{\partial \bar{S}}{\partial t}\left(x,x^{\prime },t\right)d^{n}p^{\prime }\\ & ={\left(\frac{1}{2\pi \hbar }\right)}^{n}\frac{1}{i\hbar }\int e^{\frac{i}{\hbar }\bar{S}\left(x,x^{\prime },t\right)}\bar{H}\left(x,x^{\prime },p^{\prime }\right)d^{n}p^{\prime }. \end{array}$$
Taking the limit t→0 and multiplying both sides of this equality by iℏ we finally get
$$\hat{H}\psi_{0}\left(x\right)={\left(\frac{1}{2\pi \hbar }\right)}^{n}\int e^{\frac{i}{\hbar }p\left(x-x^{\prime }\right)}\bar{H}\left(x,x^{\prime },p^{\prime },t^{\prime }\right)\psi_{0}\left(x^{\prime }\right)d^{n}p^{\prime }d^{n}x^{\prime }$$
which proves (Equation 53). □

We will call the operator $\hat{H}$ defined by (Equation 53) the Born–Jordan quantization of the Hamiltonian function *H*. That this terminology is justified is motivated below.

### 3.2. The Case of Monomials

Let us show that (Equation 53) reduces to the usual Born–Jordan quantization rule (Equation 1) when $H=x^{m}p^{\ell }$ (we are thus assuming dimension n=1). We have here
$$H\left(\tau x+\left(1-\tau \right)x^{\prime },p\right)={\left(\tau x+\left(1-\tau \right)x^{\prime }\right)}^{m}p^{\ell }$$
hence, using the binomial formula,
(55)H(τx+(1−τ)x′,p)=∑k=0mmkτk(1−τ)m−kxkpℓx′m−k.
Integrating from 0 to 1 in τ and noting that
$$\int_{0}^{1}\tau^{k}{\left(1-\tau \right)}^{m-k}d\tau =\frac{k!(m-k)!}{(m+1)!}$$
we get
$$\bar{H}\left(x,x^{\prime },p\right)=\frac{1}{m+1}\sum_{k=0}^{m}x^{k}p^{\ell }x^{\prime m-k}$$
and hence, using the definition (Equation 53) of $\hat{H}$,
$$\begin{array}{cc} \hat{H}\psi \left(x\right) & =\frac{1}{2\pi \hbar (m+1)}\sum_{k=0}^{m}\int_{-\infty }^{\infty }e^{\frac{i}{\hbar }p\left(x-x^{\prime }\right)}x^{k}p^{\ell }x^{\prime m-k}\psi \left(x^{\prime }\right)dpdx^{\prime }\\ & =\frac{x^{k}}{2\pi \hbar (m+1)}\sum_{k=0}^{m}\int_{-\infty }^{\infty }\left(\int_{-\infty }^{\infty }e^{\frac{i}{\hbar }p\left(x-x^{\prime }\right)}p^{\ell }dp\right)x^{\prime m-k}\psi \left(x^{\prime }\right)dx^{\prime }. \end{array}$$
In view of the Fourier inversion Formula (Equation 34) we have
(56)$$\frac{1}{2\pi \hbar }\int_{-\infty }^{\infty }e^{\frac{i}{\hbar }p\left(x-x^{\prime }\right)}p^{\ell }dp={\left(-i\hbar \right)}^{\ell }\delta^{\left(\ell \right)}\left(x-x^{\prime }\right)$$
so that we finally get
$$\hat{H}\psi \left(x\right)=\frac{1}{m+1}\sum_{k=0}^{m}x^{k}{\left(-i\hbar \right)}^{\ell }\frac{\partial^{\ell }}{\partial x^{\ell }}\left(x^{m-k}\psi \right),$$
which is equivalent to (Equation 1) since ${\hat{p}}^{\ell }={\left(-i\hbar \right)}^{\ell }\partial^{\ell }/\partial x^{\ell }$.

### 3.3. Physical Hamiltonians

Let us now show that the Born–Jordan quantization of a physical Hamiltonian of the type
(57)$$H=\sum_{j=1}^{n}\frac{1}{2m_{j}}{\left(p_{j}-A_{j}\left(x\right)\right)}^{2}+V\left(x\right)$$
coincide with the usual operator
(58)$$\hat{H}=\sum_{j=1}^{n}\frac{1}{2m_{j}}{\left(-i\hbar \frac{\partial }{\partial x_{j}}-A_{j}\left(x\right)\right)}^{2}+V\left(x\right)$$
obtained by Weyl quantization (the functions $A_{j}$ and *V* are assumed to be $C^{1}$). Since the quantizations of $p_{j}^{2}$, $A_{j}\left(x\right)$ and V(x) are the same in all quantization schemes (they are respectively $-\hbar^{2}\partial^{2}/\partial x_{j}^{2}$ and multiplication by $A_{j}\left(x\right)$ and V(x)), we only need to bother about the cross-products $p_{j}A\left(x\right)$. We claim that
(59)$$\hat{p_{j}A}\psi =-\frac{i\hbar }{2}\left[\frac{\partial }{\partial x_{j}}\left(A\psi \right)+A\frac{\partial \psi }{\partial x_{j}}\right],$$
from which (Equation 58) immediately follows. Let us prove (Equation 59); it is sufficient to do this in the case n=1. Denoting by $\bar{pA}$ the Born–Jordan quantization of the function pA we have
$$\bar{pA}\left(x,x^{\prime },p\right)=p\int_{0}^{1}A\left(\tau x+\left(1-\tau \right)x^{\prime }\right)d\tau =p\bar{A}\left(x,x^{\prime }\right)$$
and hence
$$\begin{array}{cc} \hat{pA}\psi \left(x\right) & =\frac{1}{2\pi \hbar }\int e^{\frac{i}{\hbar }p\left(x-x^{\prime }\right)}p\bar{A}\left(x,x^{\prime }\right)\psi \left(x^{\prime }\right)dx^{\prime }dp\\ & =\int_{-\infty }^{\infty }\left(\frac{1}{2\pi \hbar }\int_{-\infty }^{\infty }e^{\frac{i}{\hbar }p\left(x-x^{\prime }\right)}pdp\right)\bar{A}\left(x,x^{\prime }\right)\psi \left(x^{\prime }\right)dx^{\prime }. \end{array}$$
In view of (Equation 34) the expression between the square brackets is $-i\hbar \delta^{\prime }\left(x-x^{\prime }\right)$ so that
$$\begin{array}{cc} \hat{pA}\psi \left(x\right) & =-i\hbar \int_{-\infty }^{\infty }\delta^{\prime }\left(x-x^{\prime }\right)\bar{A}\left(x,x^{\prime }\right)\psi \left(x^{\prime }\right)dx^{\prime }\\ & =-i\hbar \int_{-\infty }^{\infty }\delta \left(x-x^{\prime }\right)\frac{\partial }{\partial x^{\prime }}\left(\bar{A}\left(x,x^{\prime }\right)\psi \left(x^{\prime }\right)\right)dx^{\prime }\\ & =-i\hbar \left(\frac{\partial \bar{A}}{\partial x^{\prime }}\left(x,x\right)\psi \left(x\right))+\bar{A}\left(x,x\right)\frac{\partial \psi }{\partial x^{\prime }}\left(x\right))\right) \end{array}$$
Now, by definition of $\bar{A}\left(x,x^{\prime }\right)$ we have $\bar{A}\left(x,x\right)=A\left(x\right)$ and
$$\frac{\partial \bar{A}}{\partial x^{\prime }}\left(x,x\right)=\int_{0}^{1}\left(1-\tau \right)\frac{\partial A}{\partial x}\left(x\right)d\tau =\frac{1}{2}\frac{\partial A}{\partial x}\left(x\right)$$
and hence
$$\hat{pA}\psi =-\frac{i\hbar }{2}\frac{\partial A}{\partial x}\psi -i\hbar A\frac{\partial \psi }{\partial x}$$
which is the same thing as (Equation 59).

## 4. Discussion

Both Kerner and Sutcliffe, and Cohen relied on path integral arguments which were doomed to fail because of the multiple possible choices of histories in path integration. However, it follows from our rigorous constructions that Kerner and Sutcliffe’s insight was right, even though their construction was not rigorously mathematically justified. While there is, as pointed out by Cohen [22], a great latitude in choosing the short-time propagator, thus leading to different quantizations, our argument did not make use of any path-integral argument; what we did was to propose a short-time propagator which is *exact* up to order $O(t^{2})$ (as opposed to those obtained by using midpoint rules), and to show that if one use this propagator, then one must quantize Hamiltonian functions (and in particular monomials) following the prescription proposed by Born and Jordan in the case of monomials.

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
