# Peer review of "Short-Time Propagators and the Born–Jordan Quantization Rule"

_entropy, 2018, doi:10.3390/e20110869_

Reviewer 1 Report

The aim of the author is to promote the Born-Jordan (BJ) quantisation. As a specialist of this procedure, as is proved by an impressive list of his articles and books, he considers BJ as the (most) correct one to derive a Hamiltonian operator from its classical counterpart. The adjective   "correct"  essentially refers to the multiplicity of orderings on the quantum level (i.e. the ordering problem), exemplified by the fact that  Weyl and Born-Jordan rules yield different operators for the square of the angular momentum  when the latter is expressed with Cartesian coordinates. Note that this kind of ambiguity does not appear if quantisation based on unitary irreducible representations of the group SO(3) is used instead, since the square of the angular momentum operator is just the Casimir operator which assumes a constant value for each UIR.   I expect at least a comment about this observation from the author.

In the submitted paper, the author presents another argument in favour of BJ quantisation. The rationale is based on the    Feynman path integral techniques, namely evolution operator and its kernel at short time. Among a multiplicity  of possible time averaging pairs of  two space locations, the BJ choice rests upon the averaging of  convex superposition of the pair variables in the Hamiltonian. The idea is attractive, and the final result (theorem 4) giving the action of the quantum Hamiltonian on a wave function as a kind of convolution of two Fourier transforms is certainly of great interest. In conclusion, the presented results deserve to be published in the special issue.

Besides the above remark about the angular momentum, I noticed a few misprints which should be corrected. 

1) Equation 12: "ds" should be "d\tau"

2) Footnote 2: does the author mean "rebuttal"?

3) Footnote 3: correct "explicitl", "claculate", "exlicit"

4) \psi_0 is missing in the integral term in the third equation in page 8

5) 4) \psi_0 is missing  in rhs of the  equation below Eq. (39)

6) 5) 4) \psi is missing  in rhs of the  equation below Eq. (39)

Author Response

Thank you for the kind review. The typos have been corrected.

Reviewer 2 Report

The MS by de Gosson deals with the BJ quantization rule in quantum mechanics and short time propagators. Most of the MS is already known. I would say that only Section 3 is a little bit novel. In this sense, the MS seems to me too long, it should be reduced. On the other hand, the way to present his results in terms of theorems, in my opinion, is not very appropriate.  It is a midway between physics and mathematics.

I think that MS could be eventually published after the author considers to reduce his MS and emphasizes better what is new in his work. As a recommendation, I should avoid writing the MS in terms of theorems. The same can be said without resorting to a mathematical style.

Author Response

I am mortified of having to tell you that the mathematical style of the paper is from my point of view very adequate, since this is an essentially mathematical apper, with applications to physics. The lenth is not excessive, The two other reviewers seem to be totally happy with this. Thank you anyway for your constructive suggestions.

Reviewer 3Report

The article is well written, absolutely clear and mathematically sound. 

I have no further suggestions.

Author Response

Thank you for your very kind and positive report!